# Identifying psychiatrist characteristics associated with likelihood of recommending involuntary hospitalization for patients using a novel tool to assess decision-making

Karin R. Lavie, Royce Lee, Kristen C. Jacobson ●*

Department of Psychiatry and Behavioral Neuroscience, University of Chicago, Chicago, Illinois, United States of America

* kjacobso@bsd.uchicago.edu

## Abstract

Psychiatric involuntary hospitalization (IH) rates differ across the United States (U.S.), but few studies have investigated what physician characteristics influence the decision-making process for IH. This cross-sectional survey study used the Psychiatric Involuntary Hospitalization Decision-making (PIHD) instrument, a previously validated, vignette-based tool, to measure individual psychiatrists' likelihood to admit patients involuntarily and their confidence in IH decision-making. Psychiatrists and psychiatry trainees (N = 246) from eight pre-selected academic psychiatry departments across major U.S. regions completed an online survey that included the PIHD instrument and questions on physician demographics, clinical experience, attitudes and beliefs about patient care, and level of paternalism. Results indicated that demographic factors and years of experience were not associated with likelihood of admittance or physician confidence in decision-making. Likelihood of IH admittance was higher among participants in the Northeast and Southeast. Among attending physicians, likelihood of IH admittance was higher among those with inpatient experience and lower among those with experience in psychiatric emergency services. Likelihood of admittance was also positively correlated with higher levels of paternalism and physician beliefs that IH is beneficial. Among trainees, greater worries about patient safety were associated with higher likelihood of IH admittance. In the full sample, confidence in IH decision-making was highest in the Northeast, Southeast, and Southwest, and was positively correlated with emergency psychiatry experience. Confidence in IH decision-making was associated with paternalism, but only among attending physicians. This study is one of the first to identify individual factors that may influence psychiatrist decision-making around IH in the U.S.

**Data availability statement:** A file with the statistical code and output from analyses has been uploaded as a Supporting Information file. Raw data are available at: https://knowledge.uchicago.edu/record/16834?&ln=en.

**Funding:** This research was supported by a University of Chicago Biological Sciences Division Dean's Fund Trainee Research Award (no reference number to KRL). The funder had no role in the research question, study design, data collection and statistical analyses, decision to publish, or preparation of the manuscript.

**Competing interests:** The authors have declared that no competing interests exist.

## Introduction

Psychiatric involuntary hospitalization (IH) is utilized to ensure that individuals in imminent danger of harming themselves or others receive the required level of care. Nevertheless, the decision to hospitalize is often complex, involving a balance of risks, patient autonomy, and beneficence. Currently, there are no widely accepted, clinically-grounded guidelines to standardize the decision-making process. Some psychiatrists are more conservative, valuing patient safety over autonomy, whereas other psychiatrists prioritize individual patient rights. Individual differences in how patients are treated can raise issues of quality of care. In psychiatry, this is particularly important for decisions regarding IH, given the ethical issues of removing, at least temporarily, patient autonomy in decision-making.

To date, most studies of individual differences in IH have focused on how patient characteristics, such as demographics, diagnosis, and symptom severity are associated with IH [1–5]. However, fewer studies have focused on how contextual factors and physician characteristics relate to IH decision-making [6]. Indeed, at present, there has been no systematic evaluation of whether psychiatrists across the United States (U.S.) have similar thresholds for IH or to identify the factors that influence their decisions.

Multiple studies done in Europe, Asia, Canada, and Australia show significant differences between various countries in rates of IH [7–12]. Some of these differences may be due to variations in legal policies and regulations across countries. For example, Dressing and Salize found that lower IH admission rates existed in countries that had mandatory policies of notifying relatives before admissions and in countries with obligatory legal representatives for patients [8,9]. In this study, the authors also found that danger to oneself was not a prerequisite for IH in all countries [8]. In a comparison of IH policies across Canada and Australia, Gray et al. (2010) found that in some parts of Canada, patients who have capacity cannot be admitted even when they pose harm to themselves or others, whereas in Australia, involuntary patients cannot refuse treatment [13]. There are also national differences in the number and type of experts required for IH [8]. For example, in the United Kingdom, IH requires the approval of two medical practitioners, one of whom must be a psychiatrist, as well as approval from a non-medical professional who has been specially trained as an Approved Mental Health Professional, while in Italy, the patients must have an initial assessment by a medical practitioner and by a second physician belonging to the Local Health Unit, and the Mayor of the patient's municipality must formally authorize the IH [14]. Structural factors may also play a role, as [7] found that Germany and Czech have the highest ratio of psychiatric beds to inhabitants in Europe, whereas central European countries have the highest staff shortages [7]. Cultural differences may further impact the views of IH, as one study reported that mental health care professionals in Taiwan and China were more likely to look favorably on IH than those in England, Wales, and New Zealand [15].

Within the U.S., IH rates vary widely by state and region. For example, one analysis of IH rates (per 100,000 persons) across 25 US states between 2011–2018 reported that IH rates ranged from a low of 0.00029 (Connecticut) to a high of .00966

(Florida), a 33-fold difference [16]. Accessibility to psychiatric crisis services may also affect IH rates. For example, analysis of more than 2600 patient evaluations throughout the state of Virginia revealed that fewer community-based alternatives to hospitalization, such as temporary housing or short-term crisis stabilization, were associated with IH decisions [17]. In the US, the lowest proportion of available psychiatric walk-in and crisis services is seen in the Northeast [18].

Differences in IH decision-making are also likely impacted by individual level characteristics including demographic factors, clinical experience, and physician beliefs about patient beneficence, autonomy, and related factors. For example, a study comparing physician attitudes across Taiwan, England, Wales, and New Zealand, found that females were more likely to look favorably on IH than males when the patient case involved violence against others [15]. Regarding clinician experience and training, there is evidence that patients admitted involuntarily by less experienced physicians have shorter hospitalizations [19], which may indicate that less experienced physicians more often default to IH in ambiguous situations, resulting in higher rates of unnecessary IH. The link between clinical experience and IH is further supported by the finding that psychiatrists and physicians with greater experience express less concern over being prosecuted for their IH decisions and feel more familiar with the legal policies [20]. Finally, physician attitudes and beliefs have been related to endorsement of IH, with psychiatrists less likely to endorse IH if they believed mental health providers should honor client refusal for treatment and more likely to endorse IH if they believed physicians have a responsibility to ensure basic needs of clients are met [21]. To our knowledge, there has been no quantitative study examining links between measures of physician paternalism and IH decision-making.

Efforts to understand how physician characteristics impact IH decision-making have been hampered by lack of a standardized tool to assess individual differences in likelihood of IH. Prior studies have mostly relied on aggregate measures of IH rates, which limit analyses of individual characteristics, or on physician self-reported attitudes towards IH, which may not correspond to actual IH decision-making in clinical scenarios. Recently, a novel, vignette-based instrument to assess IH decision-making was developed and validated [22]. The Psychiatric Involuntary Hospitalization Decision-making (PIHD) instrument consists of a series of hypothetical emergency psychiatry scenarios in which respondents are asked whether they would admit or discharge each patient. This instrument provides an estimate of an individual's overall likelihood of IH as well as their confidence in IH decision-making. Previous analysis of the individual PIHD vignettes revealed significant between-subject effects in average confidence levels that replicated across two different samples of psychiatrists and psychiatric trainees, suggesting that physician characteristics play a role in the IH decision-making process [22]. However, this study did not systematically explore factors that might be associated with these individual differences in IH decision-making.

Thus, the goal of the current study is to better understand the role of physician characteristics on psychiatrists' individual differences in the IH decisions using the PIHD instrument in a national sample. By conducting analyses of cross-sectional study that includes data on a wide range of physician characteristics, we hope to gain initial insights on patterns that can be used to generate hypotheses about the factors that influence individual differences in physician IH decision-making. A better understanding of individual differences could lead to greater standardization of care across practitioners, which could in turn decrease bias and increase the quality of care for all patients.

While our analyses are exploratory, we expect there will be differences in IH decision-making across region and practice setting, and that physician experience, training, and personality characteristics will be related to the likelihood of IH. Specifically, we hypothesize that factors that would contribute to higher likelihood of deciding to psychiatrically admit hypothetical patients include greater paternalism, less inpatient and emergency experience, and that likelihood of admittance will be higher among psychiatrists who are trained and/or who practice on the East Coast and Midwest, compared to the West coast.

## Materials and methods

### Ethics statement

The study protocol was reviewed by the University of Chicago Biological Sciences Division Institutional Review Board (IRB) under protocol IRB22–0606 and was considered Exempt. The IRB further determined that informed consent was

not necessary because the study was minimal risk and that any disclosure of the human subjects' responses outside the research would not reasonably place the subjects at risk of criminal or civil liability or be damaging to the subjects' financial standing, employability, educational advancement, or reputation. However, the first page of the online survey included a detailed description of the study and contact information for study staff. Respondents who continued to the next page were assumed to have provided implicit consent for participation in the study.

## Sample and procedures

This study used a cross-sectional, online survey design. Subjects were psychiatrists and psychiatric trainees aged 18 and older from eight academic psychiatry departments. Institutions were pre-selected for representation of each major U.S. region (Northeast, Southeast, Midwest, Southwest, and West). Participants completed the survey between May 2, 2022, and May 12, 2023, using a link to the online survey that was sent via email listserv from a contact at each institution. The survey took approximately 10 minutes to complete. Participants were compensated for their time with a $25 Amazon gift-card that was sent electronically.

## Measures

**The Psychiatric Involuntary Hospitalization Decision-making (PIHD) tool.** Details on the development and validation of the instrument to measure psychiatrist involuntary hospitalization decisions have been published previously [22]. In brief, the PIHD includes eight vignettes of psychiatric emergency cases. To assess participant attention and knowledge, two of the vignettes were designed to be "anchors": one a clear case for involuntary hospitalization; the other a clear case for discharge. The remaining six "complex" vignettes were more clinically difficult to elicit a greater variety of responses across psychiatrists. Following presentation of each vignette, two questions were asked: 1) "Would you admit or discharge?" coded as Yes or No; and 2) "How confident do you feel about this decision?" For the confidence question, respondents on a sliding scale of 0 ("not at all confident") to 100 ("very confident").

The primary outcome of the PIHD is the likelihood of admittance, defined as the proportion of complex vignettes in which a respondent decided to admit. The secondary outcome is the average confidence in decisions across the complex vignettes, regardless of whether the decision is to admit or discharge. One complex vignette was excluded from the calculations of the likelihood of admittance and average confidence outcomes due to lower between-subject variability of response and lower test-retest agreement [22]. Thus, the PIHD outcomes reflect responses aggregated across five different clinical scenarios.

**Participant demographic characteristics and clinical experience.** Each participant location was assigned to one of five national regions (Northeast, Southeast, Midwest, Southwest, West). All subjects were given multiple choice questions assessing age, gender, race, ethnicity, and attending versus trainee status. In accordance with guidelines used to collect data on the US population during the decennial census, separate questions were asked regarding participant racial and ethnic group. Respondents were asked to choose from a list of eight racial categories (White, Black, Southeast Asian, East Asian, Native American/Alaskan Native, Native Hawaiian/Other Pacific Islander, Middle Eastern/North African, and other racial group not listed). Multiple responses were permitted. Respondents then answered a yes or no question on whether they of Hispanic or Latino origin (ethnicity). In the US, physicians who have completed medical school do not become practicing psychiatrists until they have completed at least four years of training from an accredited institution and have obtained a state medical license to practice psychiatry, which requires passing all licensing exams and background checks required by the state they will be practicing in. Adult psychiatrists require four years of specialized training after medical school, and child psychiatrists must have at least three years of adult psychiatry training and two additional years of child psychiatry training. After completing adult or child psychiatry training, physicians can also opt to complete one- or two-year fellowships in psychiatric subspecialties. Physicians in training for adult psychiatry are referred to as residents. Physicians who are completing the two-year child psychiatry training or other specialized psychiatry training are referred

to as fellows. Physicians who have completed their training and are employed at academic medical centers are referred to as attending psychiatrists.

Participants who were trainees were asked what year of training they were in and how much time they had spent in emergency psychiatry. Participants who were attendings were asked how many years they had been practicing and what settings they currently practice in. Three variables were created to reflect inpatient settings (i.e., practicing in academic inpatient, community inpatient, state hospital, or consult-liaison settings), outpatient settings, and practicing in a psychiatric emergency setting. Participants could practice in more than one setting. Attending participants were asked what percent of time they worked as an emergency psychiatrist and whether their emergency work was in Psychiatric Emergency Services (PES), defined as a 24/7 hospital-level emergency room that is EMTALA-compliant and offers evaluation and triage of acute psychiatric conditions and crisis stabilization. All attendings were asked whether their residency training had a PES.

**Attitudes and beliefs about patient care.** Physician's views on the physician-patient relationship were assessed with a 5-item paternalism scale [23]. Participants were asked questions regarding: 1) worrying about patients after discharge, 2) using state laws to help make decisions, 3) comfort with clinical decision-making risks, and 4) belief that inpatient admission will benefit the patient. Two measures of self-perception asked participants to compare their likelihood to involuntarily admit to other psychiatrists, and their institution's likelihood to involuntarily admit to other institutions. For both questions, responses were "Less likely to admit", "Average", and "More likely to admit." Higher scores indicate greater likelihood to admit.

## Statistical analysis

Data analyses used SAS software, version 9.3 for Windows (Copyright 2002–2010, SAS Institute Inc., Cary, NC, USA). Separate univariate analyses were run for the likelihood of admittance and for the average confidence in decision-making outcomes. Comparisons among different categorical subgroups were conducted using t-test or ANOVA with post-hoc comparisons. Missing responses were omitted from statistical analyses. Low-frequency responses were either omitted (i.e., "other" gender) or combined with other categories to minimize small cell sizes. Residents and Fellows were combined into a single trainee group. The seven age group categories were collapsed into three categories (under 30; between 30 and 39; and 40 and older). Responses from the two race and ethnicity questions were combined to form two mutually exclusive groups of participants (those who identified solely as Non-Hispanic White; and those who identified as one or more of the non-White racial groups and/or identified as Hispanic or Latino). Associations with continuous or ordinal variables were tested with Pearson correlations.

## Results

### Sample characteristics

The survey was sent to an estimated 798 individuals and 246 participants had usable data, a response rate of 31%. There were 23 participants who indicated whether or not they would admit for each of the PIHD vignettes, but were missing confidence scores for three or more of the five vignettes. These participants were included in analyses of likelihood of admittance but were excluded from analyses of confidence. Table 1 presents the sample characteristics for the N = 246 with valid likelihood of admittance scores.

### PIHD outcomes

All participants correctly answered the anchor vignette with the clear admit decision and all but one participant correctly answered the anchor vignette with the clear discharge decision. Across the full sample N = 246, the average likelihood of admittance score for the five complex vignettes was 0.48 (SD = 0.25) and individual responses ranged from no admit

PLOS Mental Health

**Table 1. Sample characteristics (N = 246).**

|  | N | (%) |
|---|---|---|
| *Current Region* |  |  |
| Northeast | 40 | (16.3) |
| Southeast | 18 | (7.3) |
| Midwest | 34 | (13.8) |
| Southwest | 98 | (39.8) |
| West | 56 | (22.8) |
| *Gender* |  |  |
| Cis-Male | 112 | (43.5) |
| Cis-Female | 128 | (52.0) |
| Other Gender | 3 | (1.2) |
| No Response | 3 | (1.2) |
| *Race[a]* |  |  |
| White | 144 | (58.5) |
| Black | 8 | (3.3) |
| Southeast Asian | 33 | (13.4) |
| East Asian | 35 | (14.2) |
| Middle Eastern | 5 | (2.0) |
| Other Race | 19 | (7.7) |
| No Response | 11 | (4.5) |
| *Hispanic/Latino Ethnicity* | 35 | (14.2) |
| *Age group* |  |  |
| 18-25 | 1 | (0.4) |
| 26-29 | 75 | (30.5) |
| 30-39 | 111 | (45.1) |
| 40-49 | 36 | (14.6) |
| 50-59 | 13 | (5.3) |
| 60-69 | 7 | (2.9) |
| >70 | 2 | (0.8) |
| No Response | 1 | (0.4) |
| *Current Status* |  |  |
| Attending | 96 | (39.0) |
| Resident | 135 | (54.9) |
| Fellow | 12 | (4.9) |
| No Response | 3 | (1.2) |
| *Year in training[b]* |  |  |
| 1 | 35 | (23.8) |
| 2 | 45 | (30.6) |
| 3 | 31 | (21.1) |
| 4 | 28 | (19.1) |
| 5+ | 8 | (5.4) |
| *Years in practice[c]* |  |  |
| 0-5 | 37 | (38.5) |
| 6-10 | 22 | (22.9) |
| 11-15 | 11 | (11.5) |
| 16-20 | 12 | (12.5) |
| 21-25 | 7 | (7.3) |

*(Continued)*

**Table 1.** (Continued)

|  | N | (%) |
|---|---|---|
| >25 | 7 | (7.3) |
| *Practice Setting*[ac] |  |  |
| Emergency Psychiatry Setting | 31 | (32.3) |
| Any Outpatient Setting | 49 | (51.0) |
| *Academic outpatient | 31 | (32.3) |
| *Private practice | 16 | (16.7) |
| *Community outpatient | 5 | (5.2) |
| *Substance use treatment center | 4 | (4.2) |
| *Partial Hospitalization Program | 1 | (1.0) |
| Any Inpatient Setting | 45 | (46.9) |
| *Academic inpatient | 39 | (40.6) |
| *Consult-Liaison service | 10 | (10.4) |
| *Community inpatient | 2 | (2.1) |
| *State Hospital | 1 | (1.0) |
| Other Practice Setting | 14 | (14.6) |
| *Veterans Affairs Hospital | 8 | (8.3) |
| *County Hospital | 5 | (5.2) |
| *Correctional setting | 1 | (1.0) |
| *Percent of time as Emergency Psychiatrist*[c] |  |  |
| None | 50 | (52.1) |
| 1-20% | 20 | (20.8) |
| > 20% | 26 | (27.1) |
| *Exposure to PES during training*[c] | 54 | (56.3) |

[a]Participants were allowed to select more than one response

[b]Question asked only among the N = 147 Resident and Fellow Trainees

[c]Question asked only among the N = 96 attendings.

decisions (0) to all admit decisions (1). The average confidence score for these five vignettes among the N = 223 participants with valid data was 68.04 (SD = 14.21, range: 21–100). There was a small and non-significant correlation between participants' likelihood of admittance and the average confidence in their decisions (r = .10, N = 223, p = .14).

## Associations of PIHD outcomes with physician demographic and clinical characteristics

**Likelihood of admittance.** Fig 1 shows the average likelihood of admittance scores among the different categorical subgroups. There was a statistically significant effect of region (F [4, 241] = 6.03, p < .001; $\omega^2$ = .08 [95% CI: .01-.14]), with participants from both the Northeast and the Southeast showing statistically significantly higher likelihood of admittance than participants from the other three regions. Participants in the Northeast and Southeast did not differ from each other, nor were any of the post-hoc contrasts statistically significant between participants from the Midwest, Southwest, and West. There were no differences across gender (t = 0.75, df = 238, p = .45), racial/ethnic group (t = 1.00, df = 233, p = .32), or age group (F [2, 242] = 1.15, p = .32). Attendings who worked in inpatient settings had higher likelihood of admittance than those who did not (t = 4.58, df = 94, p < .001; Cohen's d = 0.94 [95% CI: .51- 1.36]) but there were no differences for attendings who did or did not work in outpatient (t = 1.23, df = 94, p = .22) or emergency psychiatry settings (t = 0.78,

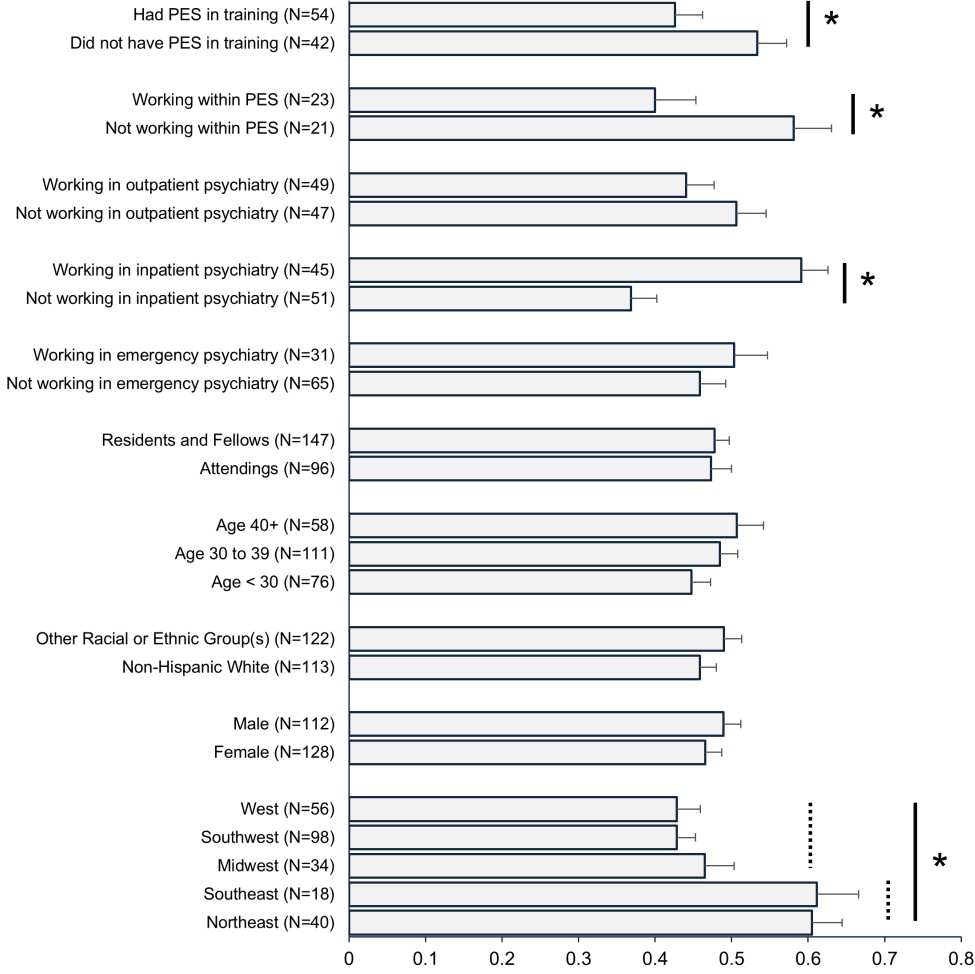

**Fig 1. Average likelihood of admittance scores among different subgroups.** Note. Error bars are standard errors for each subgroup. PES = psychiatric emergency service. Questions on practice setting and training experience with PES were asked only among attendings. Current experience with PES was asked only among attendings who indicated at least some current work in emergency psychiatry. Group differences statistically significant at p < .05 are indicated by an asterisk.

df = 94, p = .44). However, working within a PES currently (t = 2.47, df = 42, p = .018; Cohen's *d* = 0.75 [95% CI: .13-1.35]) or during training (t = 2.03, df = 94, p = .045; Cohen's *d* = 0.42 [95% CI: .01-.82]) was associated with lower likelihood of admittance. Finally, scores did not differ between attendings and trainees (t = 0.14, df = 241, p = .89). Additional correlational analyses found likelihood scores were not associated with years of practice (r = .15, N = 96, p = .15) or amount of time spent as an emergency psychiatrist (r = .03, N = 96, p = .77) among the attendings, or with post-graduate year (r = .02, N = 147, p = .84) or amount of time spent in the emergency room (r = -.07, N = 147, p = .40) among the trainees.

**Confidence in decision.** Fig 2 shows the average confidence scores across the different demographic and clinical subgroups. While post-hoc contrasts revealed that participants in the Northeast (t = 3.88, df = 218, p = .018) and Southwest (t = 2.29, df = 218, p = .02) each had statistically significantly higher confidence scores in comparison to participants in the Midwest, none of the other contrasts were significant and the overall test of regional differences was only marginally

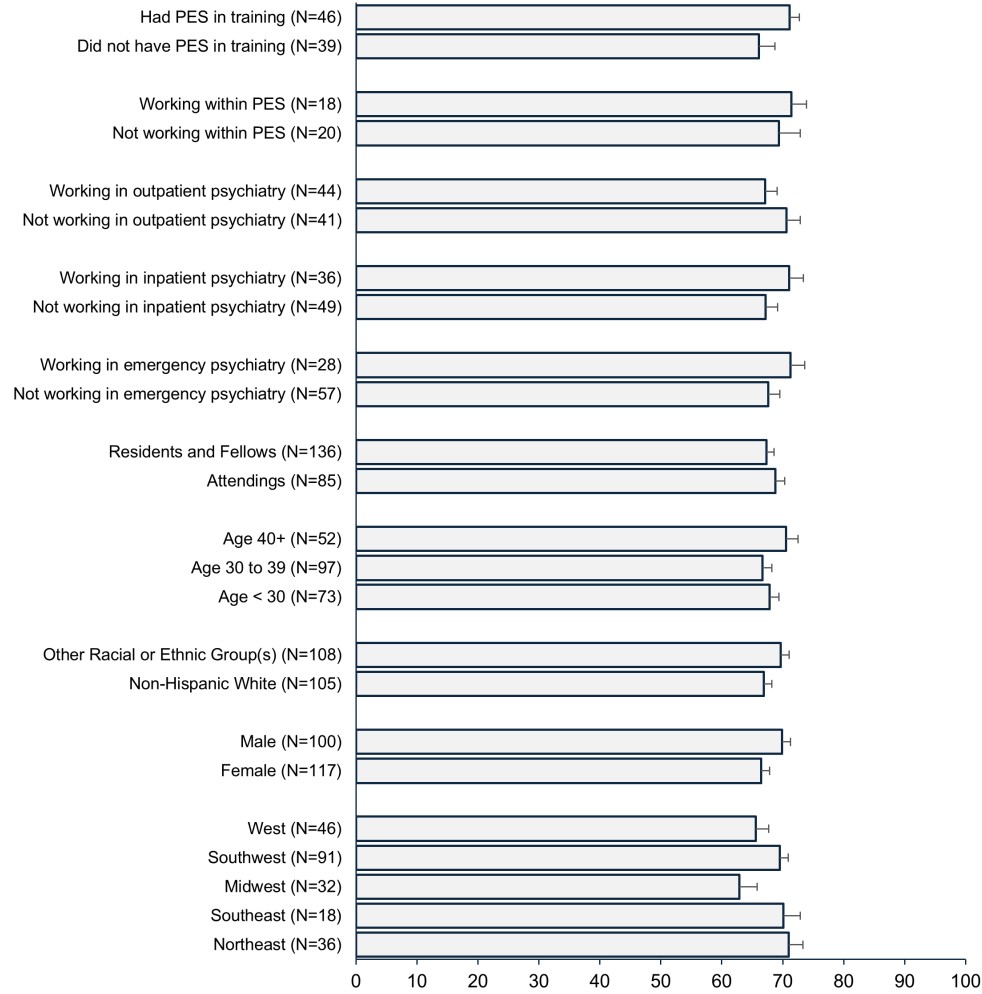

**Fig 2. Average confidence scores among different subgroups.** Note. Error bars are standard errors for each subgroup. PES = psychiatric emergency service. Questions on practice setting and training experience with PES were asked only among attendings. Current experience with PES was asked only among attendings who indicated at least some current work in emergency psychiatry. None of the group differences were statistically significant at p<.05.

significant (F[4, 218] = 2.16, p=.07; $\omega^2$=.02 [95% CI: -.02-.07]). None of the other differences across the subgroups shown in Fig 2 were statistically significant. Amount of time spent as an emergency psychiatrist was associated with greater confidence among the attendings (r=.22, 95% CI:.003-.41; N=85, p=.045) but years practicing was not (r=.09, 95% CI: -.13-.30; N=85, p=.42). For trainees, post-graduate year was associated with greater confidence (r=.32, 95% CI:.16-.47; N=136, p<.001), and there was a trend towards greater confidence with more time spent in the emergency room (r=.16, 95% CI: -.01-.32; N=136, p=.065).

**Associations of PIHD outcomes with physician personality, attitudes, and beliefs.** Table 2 shows correlations between the two PIHD outcomes with measures of physician personality, attitudes, and beliefs. Using laws in decision-making was not statistically significantly related to either PIHD outcome. Higher paternalism was related to both higher likelihood of admittance as well as greater confidence in decision-making, but only among the attendings. Likewise, stronger belief that IH has benefits for patients was associated with both higher likelihood of admittance as well as

**Table 2. Correlations (95% CI) between PIHD outcomes with physician personality and beliefs.**

| | Likelihood of Admittance | | | Confidence in decision | | |
|---|---|---|---|---|---|---|
| | Full Sample<br>N = 240–241 | Attendings<br>N = 94–95 | Trainees<br>N = 145–146 | Full Sample<br>N = 219–219 | Attendings<br>N = 84 | Trainees<br>N = 134–135 |
| Paternalism | **.14**<br>**(.02;.27)** | **.26**<br>**(.06;.44)** | .06<br>(-.10;.22) | **.15**<br>**.02;.28)** | **.23**<br>**(.01;.42)** | .13<br>(-.04;.29) |
| Worry about patients | .09<br>(-.04;.21) | -.07<br>(-.27;.13) | **.20**<br>**(.05;.36)** | **-.15**<br>**(-.28;-.02)** | **-.26**<br>**(-.44;-.04)** | -.10<br>(-.26;.07) |
| Use laws in<br>decision-making | -.04<br>(-.17;.08) | .11<br>(-.09;.30) | -.14<br>(-.30;.02) | .08<br>(-.05;.21) | .00<br>(-.22;.21) | .11<br>(-.06;.27) |
| Comfort with risk in decision-making | .04<br>(-.09;.16) | **.23**<br>**(.02;.41)** | -.11<br>(-.27;.05) | **.39**<br>**(.27;.49)** | **.39**<br>**(.19;.56)** | **.38**<br>**(.22;.51)** |
| IH has benefits for<br>patients | **.19**<br>**(.06;.31)** | **.32**<br>**(.12;.49)** | .08<br>(-.09;.24) | **.16**<br>**(.02;.28)** | **.36**<br>**(.16;.54)** | .01<br>(-.15;.18) |
| Admit compared to other physicians | **.21**<br>**(.09;.33)** | **.27**<br>**(.07;.45)** | **.17**<br>**(.01;.33)** | .04<br>(-.10;.17) | .00<br>(-.21;.22) | .04<br>(-.12;.21) |
| Admit compared to other institutions | **.15**<br>**(.02;.27)** | .17<br>(-.03;.36) | .13<br>(-.03;.29) | -.06<br>(-.20;.07) | .05<br>(-.17;.26) | -.14<br>(-.30;.03) |

Notes. PIHD = Psychiatric Involuntary Hospitalization Decision-Making. CI = Confidence interval.

Statistically significant associations (p < .05) are indicated in bold.

greater confidence in decision-making among the attendings. More comfort with risk in decision-making was statistically significantly correlated with higher likelihood of admittance among attendings and was associated with greater confidence among both attendings and trainees. Worrying about patient safety was associated with greater likelihood of admittance among trainees and with lower confidence in decisions among attendings. Finally, both attending and trainee perceptions of their own IH rates relative to other physicians were statistically significantly correlated with likelihood of admittance on the PIHD. When attendings and trainees were combined, perceptions of rates of IH among their institution relative to other institutions were also associated with greater likelihood of admittance, but correlations were not statistically significant in either subgroup. The two self-perception measures were not statistically significantly associated with confidence in decision-making in either participant subgroup.

## Discussion

This study examined how individual physician characteristics and attitudes affect IH decision-making in a sample of U.S. psychiatrists. A novel feature of the study is that it used a standardized, vignette-based instrument to assess both likelihood of IH and confidence in decision-making. This study adds to a limited body of work understanding individual differences in IH decision-making, most of which has been done outside the U.S. Although our exploratory approach requires replication in other samples, we found that several factors were associated with IH decision-making, but patterns differed for individual differences in likelihood of admittance versus confidence in decision. Results from this study are important because they signal that there is significant variation in practice across psychiatrists when faced with the same clinical scenario.

None of the demographic characteristics (physician gender, race/ethnicity, or age) were associated with the likelihood of admittance or confidence. Practice region was not associated with confidence but was associated with the likelihood to admit, with higher likelihood of admittance among participants from the Northeast and Southeast compared with other regions. Local laws and structural barriers may partially account for these regional differences by raising the threshold for when someone can be admitted involuntarily. For example, in California, a court order is required for IH, whereas in New York, it is not [24]. Additionally, although most states require the criteria of danger to self and others, Georgia only requires

the criterion of being mentally ill to be admitted involuntarily [24]. In the U.S., 22 states have adopted the 72-hour hold, whereas other states limit holds to 24/48 hours [24]. Previous research indicates 72-hour holds lead to decreased admission rates, perhaps by allowing more time for patient stabilization [25].

Clinical experience and training also play a role in decision-making for IH. Attending psychiatrists with inpatient psychiatric experience had higher likelihood of recommending IH, perhaps because they see patients already admitted. In contrast, attending psychiatrists with PES experience had a lower likelihood to admit. This coincides with previous data on PES that demonstrates lower psychiatric hospitalization rates [26]. The more time a participant spent in the emergency room, the higher their confidence in their decision-making, although there was no association with the likelihood of admittance. Postgraduate year was also associated with confidence, which could be due to greater emergency room exposure. However, the likelihood of admittance was not associated with postgraduate year among trainees or with years of practice among attending psychiatrists; there was also no difference in confidence or likelihood of admittance between attendings and trainees. This suggests that psychiatrists' experience with specific patient populations and specific clinical settings plays a greater role in IH decision-making than general experience in psychiatry.

There were several individual characteristics associated with physician's likelihood to admit and their overall confidence of decision-making. Patterns differed for attendings and trainees. Worrying about patient safety after discharge was associated with likelihood of admittance among trainees but not attendings. If someone is worried about patient safety, admitting them to care may reduce that concern. Because trainees have less experience than attendings, their level of worry may have a greater impact on IH decision-making. On the other hand, belief that IH benefits patients was correlated with a higher likelihood of admittance and with greater confidence among attendings, but not trainees. Physicians who have experienced more patient benefits with IH may feel more favorable towards it due to affective reasoning biases. Among the attendings (but not trainees), higher paternalism scores and greater comfort with risk in clinical decision-making were also associated with a higher likelihood of admittance and higher confidence levels. These findings suggest that when physicians prioritize their judgement over a patient's autonomy, they have more confidence and are more likely to admit.

Finally, among the whole sample, there were statistically significant correlations between the self-perception measures and the likelihood of admittance. This provides additional validation of the PIHD as a standardized tool that can be used as a proxy for measuring an individual's likelihood of involuntary admittance.

## Limitations

One limitation of this study is that only five states were represented, which prohibited a detailed comparison of how local laws and structural factors impact IH decision-making. Only academic institutions were included, which limits generalizability. Another limitation is that the study did not control for differences in resources across institutions, including the availability of psychiatric inpatient beds and number of hospital staff. Likewise, one institution may have greater access to mental health funding or differences in patient population that affect cultural practices in IH decision-making. Future studies could expand this work to a wider variety of institutions to test whether the likelihood to admit is related to patient characteristics, access to resources, and other institutional factors. Finally, this study solely focused on psychiatrist decision-making, even though many states allow other mental health practitioners to involuntarily hospitalize patients. Future research is needed to determine whether factors associated with IH decision-making as measured by the PIHD replicate among other providers who make IH decisions.

## Conclusions

This study identified factors, including practice region, specific clinical experience, and individual beliefs and personality characteristics, that are correlated with differences in psychiatric involuntary hospitalization decision-making and confidence among U.S. psychiatrists.

# Supporting information

**S1 File. Statistical code and output.** The supplemental S1 File contains summaries of aggregated data, output from all statistical analyses reported in the paper, and the underlying statistical code from SAS.
(PDF)

# Author contributions

**Conceptualization:** Karin R. Lavie, Royce Lee, Kristen C Jacobson.

**Data curation:** Karin R. Lavie, Kristen C Jacobson.

**Formal analysis:** Kristen C Jacobson.

**Funding acquisition:** Karin R. Lavie.

**Investigation:** Karin R. Lavie, Kristen C Jacobson.

**Project administration:** Karin R. Lavie.

**Supervision:** Royce Lee, Kristen C Jacobson.

**Visualization:** Kristen C Jacobson.

**Writing – original draft:** Karin R. Lavie, Kristen C Jacobson.

**Writing – review & editing:** Karin R. Lavie, Royce Lee, Kristen C Jacobson.

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
