## [Decision Letter · Decision Letter 0]

16 Dec 2025

PMEN-D-25-00459

Identifying psychiatrist characteristics associated with likelihood of recommending involuntary hospitalization for patients using a novel tool to assess decision-making

PLOS Mental Health

Dear Dr. ,

Thank you for submitting your manuscript to PLOS Mental Health. After careful consideration, we feel that it has merit but does not fully meet PLOS Mental Health’s publication criteria as it currently stands. Therefore, we invite you to submit a revised version of the manuscript that addresses the points raised during the review process.

We look forward to receiving your revised manuscript.

Kind regards,

Lorenzo Pelizza, Ph.D.

Academic Editor

PLOS Mental Health

Journal Requirements:

1. Your current Financial Disclosure states, “This work was supported by an internal institutional Dean's Fund award to the first author.”. However, your funding information on the submission form indicates that you did not receive funding. Please indicate by return email the full and correct funding information for your study and confirm the order in which funding contributions should appear. Please be sure to indicate whether the funders played any role in the study design, data collection and analysis, decision to publish, or preparation of the manuscript.

2. Please provide separate figure files in .tif or .eps format.

https://journals.plos.org/mentalhealth/s/figures

https://journals.plos.org/mentalhealth/s/figures#loc-file-requirements

3. Please insert an Ethics Statement at the beginning of your Methods section, under a subheading 'Ethics Statement'. It must include:

1) The name(s) of the Institutional Review Board(s) or Ethics Committee(s)

2) The approval number(s), or a statement that approval was granted by the named board(s)

3) (for human participants/donors) - A statement that formal consent was obtained (must state whether verbal/written) OR the reason consent was not obtained (e.g. anonymity). NOTE: If child participants, the statement must declare that formal consent was obtained from the parent/guardian.

4. We do not publish any copyright or trademark symbols that usually accompany proprietary names, eg (R), (C), or TM  (e.g. next to drug or reagent names). Please remove all instances of trademark/copyright symbols throughout the text, including © on page 8.

Additional Editor Comments:

Dear Authors,

a decision was made about your paper: major revision. I attached the reviewers' comments below.

Please, answer to all suggestions adequately.

Best regards.

Lorenzo Pelizza.

REVIEWER 1

Reviewer Recommendation Term: Major Revision

This is an important research topic and the study was well executed within available resource constraints. The authors do not step outside the limitations of cross-sectional online surveys. The writing is mostly clear, but the manuscript needs further work to do this justice before publication in my view.

The research questions are not very precise and rather exploratory: "to better understand the role of physician characteristics on psychiatrists' individual differences in the IH decisions using the PIHD instrument in a national sample". Rather than a clear hypothesis or set of questions, the authors say "We expect there will be differences" across effect modifiers / subgroups (region, practice setting) [note there is no statistical power calculation to support subgroup analysis and I suspect the study is under-powered for these].

Then "factors that would contribute...include" permits a range of factors (i.e. predictor variables or exposures) beyond those explicitly stated.

It is better in my opinion to focus on one or few risk factors / exposures / independent variables of primary interest and then treat other variables as covariates/confounders - subgroups as effect modifiers / interaction terms only if theoretically justified and sample size is large enough. It is difficult to recruit using online surveys and achieve large sample sizes (I have attempted this myself) so I empathise with the sample size issue, but I doubt there is enough statistical power for subgroup analyses here (unless you can convince me with a formal statistical power calculation).

Going forward I recommend to pre-register very specific research questions (and any subgroups proposed) before analysis, on Open Science Framework (OSF) or similar. This is not a requirement for publication in PLOS Mental Health, but will improve precision and help convince peers that there were more precise questions beyond general understanding, expecting differences of some kind, and permitted a wide range of predictors to be "significant" (see point below - avoid p value thresholds).

Many reviewers and readers will use critical appraisal checklists to review manuscripts (https://casp-uk.net/casp-checklists/CASP-checklist-cross-sectional-study-2024.pdf), and when reading this version, I nearly ticked "Can't tell" for the first question "Did the study address a clearly focused issue?". At which point, many would stop or have been trained to not continue reading any paper unless the answer is "yes" to this first question. I continued because I think the research has been conducted well and the aims/objectives are clear - they are just not clear enough in the writing yet (particularly at the end of the introduction). The authors understand what they have done, and I think I understand, but had to re-read several parts of the manuscript in order to reach or check my understanding, which is not ideal - it should be rewritten so that it cannot be misunderstood and readers don't have to check back to the introduction to remind themselves what the specific research questions were.

CASP 7. What is the main result - I can't tell yet, but this is not a flaw in the analysis (it just needs further work on the writing)

CASP 9. Is there a clear statement of findings - I can't tell yet.

Other questions are "yes" in my view - so this can all be addressed in a revision. I don't want to sound discouraging, there is lots of good work here, and it will be suitable for publication in my view when rewritten.

The take-home message is not clear enough yet. I struggled to get a sense of what this would add to our understanding - there are lots of interesting insights, but the writing doesn't tell the story currently. This is good work, but you need to hold our hands and connect the introduction, methods, results, conclusions together with a bolder narrative arc. This does matter in scientific writing, and will strengthen any revision. What is the message in 12 words? See https://www.sciencedirect.com/science/article/pii/S2666636725010668 and attachment which I find useful when writing papers - think of the message as underpinning the introduction, methods, results and discussion each time (and even the title - which can often be a simple statement of the paper's message). It connects everything together into a narrative and readers are more likely to read/remember/cite the paper if its message is clear.

The data is not yet fully available. It is not clear where readers can find the data.

Other comments and suggestions:

L41 For readers outside US, can you provide some detail on the extent to which state vs federal laws (between and within) vary? Variation in psychiatrists' decisions is the focus, but presumably there is variation across states to consider. L50 onwards has a good summary of international variation (this could be a review paper in its own right!) but for this paper, the between and within state variation is more relevant I think.

In case interesting/relevant, in the United Kingdom, sectioning (forced hospitalization) requires two psychiatrists, and an AMHP (social worker, nurse, occupational therapist or psychologist). The AMHP could prevent hospitalisation against the recommendation of the two psychiatrists. https://www.legislation.gov.uk/ukpga/2007/12/notes/division/6/1/2?view=plain

L67 This is confusing because four locations are compared, two with relative and two with absolute measures. If possible, report the absolute for all four (or relative for all four). It reads awkwardly if these difference lenses are mixed.

L68 This is a crucial point - without crisis support, severity of psychosis might be much higher at the point hospitalization is required. Consider expanding this. When under-resourced, are psychiatrists more likely to recommend hospitalization because severity of episode at the point of presentation is higher than it might be if crisis and other support had been available earlier? Is this a confounding factor - the decisions might be quite different if presenting with less severe symptoms owing to other sources of support. Obviously it will depend - can you strengthen the argument that the vignettes fully capture scenarios where things have gotten so serious that hospital admission is almost certainly needed, vs. someone has found earlier, alternative support, and might not need it?

L81 Again, I think worth mentioning that some countries require two psychiatrists to decide, which may lower concern (particularly if one is more experienced than the other).

L110 It is not an essential requirement, but consider pre-registering your research question on Open Science Framework (OSF) or similar with hypotheses, subgroups etc.

L115 Report N and % psychiatrists vs trainees.

L117 Report ethical approval number/reference

L123 Compensated how (was it identifiable data)? How much for how long?

L148-157 Did you consider possible differences between men/women?

L173 What is meant by race vs. ethnicity? Why not "ethnic group"

L182 Typo "for a response rate" - presumably "a response rate"

L190 Now reaching Table 1, gender differences are included. But why not mentioned above with other characteristics? Small cell counts should be treated with caution, because these cannot be meaningfully analysed as categories (n = 3). Be cautious for statistical disclosure concerns prior to publication (e.g. in the UK, Office of National Statistics say no cells <10 people should be published even if risk of identification is low). I would report that small groups were included/eligible/represented, but excluded for practical/statistical reasons.

For readers outside the US, explain Current Status (Attending, Resident, Fellow) and what is "trainee" vs. "psychiatrist"? Is there a direct mapping onto year in training, if they were cross-tabulated? Surely they are very highly correlated? Do you need both?

L167 Is "Statistical analysis" better here as a heading?

L198 The analytic sample has changed to N = 246 (previously 248). It is clear why, but for simplicity I think it is better to define analytic sample as those with available data and use the same N throughout, if missing data proportion is small and completely at random. If only 2 people are lost, this is immaterial. Report as excluded from the off.

L271 I recommend to avoid p values and use 95% confidence intervals (here, and throughout). p values are sensitive to sample size. CIs convey the same information, and additionally the precision of the estimates. The effect sizes (R, beta etc.) convey the effect size.

L277 "This study explored factors" is not a good way to start a discussion section of a scientific paper. Start with a bold reminder of the message.

L346 There is unlikely to be a conflict here, but it is recommended to include a statement that the funder (even if at the Institution) had no role in the research question / study conception , analysis, decision to publish etc.

Figure 1 and 2 - these are not referenced in the text. What is their purpose, when we have tables already? Are the bars standard errors or confidence intervals? I don't think the figure is needed, the table can include all the information here.

REVIEWER 2

Reviewer Recommendation Term: accept with comments

50-54: valid point made about variations in legal policies and regulations across countries, with a relevant reference given

54: “The study authors also found that dangers to oneself was not a prerequisite….”

Suggested edit: “In this study, the authors also found that… / the authors in the study also found that…”

59: “There are also national differences in the number and type of experts required for certification.”

Comment: Please provide additional information into what "certification" entails

68-70: Accessibility is an important point for this paper, can benefit from additional discussion

159: “...was used to assess physician’s views on the….”

Suggested edit: “physicians’ views on the…”

311-312: “If someone is worried about patient safety, admitting them to care would reduce that concern.”

Suggested edit: “could”/”may” in place of “would”

329-334 Limitations Section

Comment: Something to add here would be availability of inpatient beds / availability of hospital staff

Figures are a little blurry on paper prior to clicking on access/download link. Is there any way the image can be more clear?

Overall, very well written paper

Reviewers' comments:

Reviewer's Responses to Questions

**Comments to the Author**

1. Does this manuscript meet PLOS Mental Health’s publication criteria? Is the manuscript technically sound, and do the data support the conclusions? The manuscript must describe methodologically and ethically rigorous research with conclusions that are appropriately drawn based on the data presented.? Is the manuscript technically sound, and do the data support the conclusions? The manuscript must describe methodologically and ethically rigorous research with conclusions that are appropriately drawn based on the data presented.

Reviewer #1: Yes

Reviewer #2: Yes

2. Has the statistical analysis been performed appropriately and rigorously?

Reviewer #1: I don't know

Reviewer #2: Yes

3. Have the authors made all data underlying the findings in their manuscript fully available (please refer to the Data Availability Statement at the start of the manuscript PDF file)?

The PLOS Data policy requires authors to make all data underlying the findings described in their manuscript fully available without restriction, with rare exception. The data should be provided as part of the manuscript or its supporting information, or deposited to a public repository. For example, in addition to summary statistics, the data points behind means, medians and variance measures should be available. If there are restrictions on publicly sharing data—e.g. participant privacy or use of data from a third party—those must be specified.requires authors to make all data underlying the findings described in their manuscript fully available without restriction, with rare exception. The data should be provided as part of the manuscript or its supporting information, or deposited to a public repository. For example, in addition to summary statistics, the data points behind means, medians and variance measures should be available. If there are restrictions on publicly sharing data—e.g. participant privacy or use of data from a third party—those must be specified.

Reviewer #1: No

Reviewer #2: Yes

4. Is the manuscript presented in an intelligible fashion and written in standard English?

Reviewer #1: Yes

Reviewer #2: Yes

Reviewer #1: This is an important research topic and the study was well executed within available resource constraints. The authors do not step outside the limitations of cross-sectional online surveys. The writing is mostly clear, but the manuscript needs further work to do this justice before publication in my view.

The research questions are not very precise and rather exploratory: "to better understand the role of physician characteristics on psychiatrists' individual differences in the IH decisions using the PIHD instrument in a national sample". Rather than a clear hypothesis or set of questions, the authors say "We expect there will be differences" across effect modifiers / subgroups (region, practice setting) [note there is no statistical power calculation to support subgroup analysis and I suspect the study is under-powered for these].

Then "factors that would contribute...include" permits a range of factors (i.e. predictor variables or exposures) beyond those explicitly stated.

It is better in my opinion to focus on one or few risk factors / exposures / independent variables of primary interest and then treat other variables as covariates/confounders - subgroups as effect modifiers / interaction terms only if theoretically justified and sample size is large enough. It is difficult to recruit using online surveys and achieve large sample sizes (I have attempted this myself) so I empathise with the sample size issue, but I doubt there is enough statistical power for subgroup analyses here (unless you can convince me with a formal statistical power calculation).

Going forward I recommend to pre-register very specific research questions (and any subgroups proposed) before analysis, on Open Science Framework (OSF) or similar. This is not a requirement for publication in PLOS Mental Health, but will improve precision and help convince peers that there were more precise questions beyond general understanding, expecting differences of some kind, and permitted a wide range of predictors to be "significant" (see point below - avoid p value thresholds).

Many reviewers and readers will use critical appraisal checklists to review manuscripts (https://casp-uk.net/casp-checklists/CASP-checklist-cross-sectional-study-2024.pdf), and when reading this version, I nearly ticked "Can't tell" for the first question "Did the study address a clearly focused issue?". At which point, many would stop or have been trained to not continue reading any paper unless the answer is "yes" to this first question. I continued because I think the research has been conducted well and the aims/objectives are clear - they are just not clear enough in the writing yet (particularly at the end of the introduction). The authors understand what they have done, and I think I understand, but had to re-read several parts of the manuscript in order to reach or check my understanding, which is not ideal - it should be rewritten so that it cannot be misunderstood and readers don't have to check back to the introduction to remind themselves what the specific research questions were.

CASP 7. What is the main result - I can't tell yet, but this is not a flaw in the analysis (it just needs further work on the writing)

CASP 9. Is there a clear statement of findings - I can't tell yet.

Other questions are "yes" in my view - so this can all be addressed in a revision. I don't want to sound discouraging, there is lots of good work here, and it will be suitable for publication in my view when rewritten.

The take-home message is not clear enough yet. I struggled to get a sense of what this would add to our understanding - there are lots of interesting insights, but the writing doesn't tell the story currently. This is good work, but you need to hold our hands and connect the introduction, methods, results, conclusions together with a bolder narrative arc. This does matter in scientific writing, and will strengthen any revision. What is the message in 12 words? See https://www.sciencedirect.com/science/article/pii/S2666636725010668 and attachment which I find useful when writing papers - think of the message as underpinning the introduction, methods, results and discussion each time (and even the title - which can often be a simple statement of the paper's message). It connects everything together into a narrative and readers are more likely to read/remember/cite the paper if its message is clear.

The data is not yet fully available. It is not clear where readers can find the data.

Other comments and suggestions:

L41 For readers outside US, can you provide some detail on the extent to which state vs federal laws (between and within) vary? Variation in psychiatrists' decisions is the focus, but presumably there is variation across states to consider. L50 onwards has a good summary of international variation (this could be a review paper in its own right!) but for this paper, the between and within state variation is more relevant I think.

In case interesting/relevant, in the United Kingdom, sectioning (forced hospitalization) requires two psychiatrists, and an AMHP (social worker, nurse, occupational therapist or psychologist). The AMHP could prevent hospitalisation against the recommendation of the two psychiatrists. https://www.legislation.gov.uk/ukpga/2007/12/notes/division/6/1/2?view=plain

L67 This is confusing because four locations are compared, two with relative and two with absolute measures. If possible, report the absolute for all four (or relative for all four). It reads awkwardly if these difference lenses are mixed.

L68 This is a crucial point - without crisis support, severity of psychosis might be much higher at the point hospitalization is required. Consider expanding this. When under-resourced, are psychiatrists more likely to recommend hospitalization because severity of episode at the point of presentation is higher than it might be if crisis and other support had been available earlier? Is this a confounding factor - the decisions might be quite different if presenting with less severe symptoms owing to other sources of support. Obviously it will depend - can you strengthen the argument that the vignettes fully capture scenarios where things have gotten so serious that hospital admission is almost certainly needed, vs. someone has found earlier, alternative support, and might not need it?

L81 Again, I think worth mentioning that some countries require two psychiatrists to decide, which may lower concern (particularly if one is more experienced than the other).

L110 It is not an essential requirement, but consider pre-registering your research question on Open Science Framework (OSF) or similar with hypotheses, subgroups etc.

L115 Report N and % psychiatrists vs trainees.

L117 Report ethical approval number/reference

L123 Compensated how (was it identifiable data)? How much for how long?

L148-157 Did you consider possible differences between men/women?

L173 What is meant by race vs. ethnicity? Why not "ethnic group"

L182 Typo "for a response rate" - presumably "a response rate"

L190 Now reaching Table 1, gender differences are included. But why not mentioned above with other characteristics? Small cell counts should be treated with caution, because these cannot be meaningfully analysed as categories (n = 3). Be cautious for statistical disclosure concerns prior to publication (e.g. in the UK, Office of National Statistics say no cells <10 people should be published even if risk of identification is low). I would report that small groups were included/eligible/represented, but excluded for practical/statistical reasons.

For readers outside the US, explain Current Status (Attending, Resident, Fellow) and what is "trainee" vs. "psychiatrist"? Is there a direct mapping onto year in training, if they were cross-tabulated? Surely they are very highly correlated? Do you need both?

L167 Is "Statistical analysis" better here as a heading?

L198 The analytic sample has changed to N = 246 (previously 248). It is clear why, but for simplicity I think it is better to define analytic sample as those with available data and use the same N throughout, if missing data proportion is small and completely at random. If only 2 people are lost, this is immaterial. Report as excluded from the off.

L271 I recommend to avoid p values and use 95% confidence intervals (here, and throughout). p values are sensitive to sample size. CIs convey the same information, and additionally the precision of the estimates. The effect sizes (R, beta etc.) convey the effect size.

L277 "This study explored factors" is not a good way to start a discussion section of a scientific paper. Start with a bold reminder of the message.

L346 There is unlikely to be a conflict here, but it is recommended to include a statement that the funder (even if at the Institution) had no role in the research question / study conception , analysis, decision to publish etc.

Figure 1 and 2 - these are not referenced in the text. What is their purpose, when we have tables already? Are the bars standard errors or confidence intervals? I don't think the figure is needed, the table can include all the information here.

Reviewer #2: 50-54: valid point made about variations in legal policies and regulations across countries, with a relevant reference given

54: “The study authors also found that dangers to oneself was not a prerequisite….”

Suggested edit: “In this study, the authors also found that… / the authors in the study also found that…”

59: “There are also national differences in the number and type of experts required for certification.”

Comment: Please provide additional information into what "certification" entails

68-70: Accessibility is an important point for this paper, can benefit from additional discussion

159: “...was used to assess physician’s views on the….”

Suggested edit: “physicians’ views on the…”

311-312: “If someone is worried about patient safety, admitting them to care would reduce that concern.”

Suggested edit: “could”/”may” in place of “would”

329-334 Limitations Section

Comment: Something to add here would be availability of inpatient beds / availability of hospital staff

Figures are a little blurry on paper prior to clicking on access/download link. Is there any way the image can be more clear?

Overall, very well written paper

**Do you want your identity to be public for this peer review?** For information about this choice, including consent withdrawal, please see our Privacy Policy..

Reviewer #1: **Yes:** Gareth Hagger-JohnsonGareth Hagger-JohnsonGareth Hagger-JohnsonGareth Hagger-Johnson

Reviewer #2: No

---

## [Decision Letter · Decision Letter 1]

25 Mar 2026

Identifying psychiatrist characteristics associated with likelihood of recommending involuntary hospitalization for patients using a novel tool to assess decision-making

PMEN-D-25-00459R1

Dear Dr Jacobson,

We are pleased to inform you that your manuscript 'Identifying psychiatrist characteristics associated with likelihood of recommending involuntary hospitalization for patients using a novel tool to assess decision-making' has been provisionally accepted for publication in PLOS Mental Health.

Best regards,

Lorenzo Pelizza, Ph.D.

Academic Editor

PLOS Mental Health

Reviewer Comments (if any, and for reference):

Reviewer's Responses to Questions

**Comments to the Author**

Reviewer #3: All comments have been addressed

publication criteria? Is the manuscript technically sound, and do the data support the conclusions? The manuscript must describe methodologically and ethically rigorous research with conclusions that are appropriately drawn based on the data presented.? Is the manuscript technically sound, and do the data support the conclusions? The manuscript must describe methodologically and ethically rigorous research with conclusions that are appropriately drawn based on the data presented.

Reviewer #3: Yes

3. Has the statistical analysis been performed appropriately and rigorously?

Reviewer #3: Yes

4. Have the authors made all data underlying the findings in their manuscript fully available (please refer to the Data Availability Statement at the start of the manuscript PDF file)?

The PLOS Data policy requires authors to make all data underlying the findings described in their manuscript fully available without restriction, with rare exception. The data should be provided as part of the manuscript or its supporting information, or deposited to a public repository. For example, in addition to summary statistics, the data points behind means, medians and variance measures should be available. If there are restrictions on publicly sharing data—e.g. participant privacy or use of data from a third party—those must be specified.requires authors to make all data underlying the findings described in their manuscript fully available without restriction, with rare exception. The data should be provided as part of the manuscript or its supporting information, or deposited to a public repository. For example, in addition to summary statistics, the data points behind means, medians and variance measures should be available. If there are restrictions on publicly sharing data—e.g. participant privacy or use of data from a third party—those must be specified.

Reviewer #3: Yes

5. Is the manuscript presented in an intelligible fashion and written in standard English?

Reviewer #3: Yes

Reviewer #3: I have no additional comments

**Do you want your identity to be public for this peer review?** For information about this choice, including consent withdrawal, please see our Privacy Policy..

Reviewer #3: **Yes:** Mah Wasi AsombangMah Wasi AsombangMah Wasi AsombangMah Wasi Asombang
